# MERLIN: a novel BRET-based proximity biosensor for studying mitochondria–ER contact sites

Vanessa Hertlein[1,]*, Hector Flores-Romero[1,]*, Kushal K Das[1], Sebastian Fischer[2], Michael Heunemann[3], Maria Calleja-Felipe[4], Shira Knafo[4,5,6], Katharina Hipp[7], Klaus Harter[3], Julia C Fitzgerald[8], Ana J García-Sáez[1]

The contacts between the ER and mitochondria play a key role in cellular functions such as the exchange of lipids and calcium between both organelles, as well as in apoptosis and autophagy signaling. The molecular architecture and spatiotemporal regulation of these distinct contact regions remain obscure and there is a need for new tools that enable tackling these questions. Here, we present a new bioluminescence resonance energy transfer–based biosensor for the quantitative analysis of distances between the ER and mitochondria that we call MERLIN (Mitochondria–ER Length Indicator Nanosensor). The main advantages of MERLIN compared with available alternatives are that it does not rely on the formation of artificial physical links between the two organelles, which could lead to artifacts, and that it allows to study contact site reversibility and dynamics. We show the applicability of MERLIN by characterizing the role of the mitochondrial dynamics machinery on the contacts of this organelle with the ER.

## Introduction

Membrane contact sites are distinct, juxtaposed regions between heterotypic membranous organelles that are physically associated via tethers of protein and lipid nature. They play a critical role in inter-organelle communication, including non-vesicular transport of small molecules, such as lipids and ions, as well as signaling and metabolic pathways. During the last decade, our understanding of the functional relevance and architecture of membrane contact sites has improved dramatically and revealed an unanticipated complexity that remains poorly understood (Bohnert & Schuldiner, 2018).

Some of the best characterized membrane contact sites correspond to the domains that mediate the physical interaction between the ER and mitochondria, which are known as mitochondria–ER membrane contacts (MERCs) or mitochondria-associated membranes (Poston et al, 2013). They influence multiple cellular functions such as the coordination of calcium signaling (Rosario Rizzuto, 1998), lipid biosynthesis and transfer (Vance, 1990; Voelker, 2005), the regulation of apoptosis (Pinton et al, 2008; Grimm, 2012), autophagy (Hailey et al, 2010; Hamasaki et al, 2013), and mitochondrial dynamics (Friedman et al, 2011). Furthermore, there is evidence that MERC morphology is altered in several human diseases, including neurodegenerative diseases (Area-Gomez et al, 2012) and cancer (Carlotta Giorgi et al, 2010), which makes them a promising target for biomedical applications.

Only small areas of ~5–20% of the ER surface are in close apposition to the mitochondria, where the inter-organelle distance ranges between 10 and 30 nm, as shown by high resolution and three-dimensional reconstructions of EM studies (Csordas et al, 2006; Vance, 2014). In yeast, MERCs are kept together thanks to a complex of known composition called ERMES (Kornmann et al, 2009). However, the molecular architecture of the complexes responsible for MERCs in mammals is more complex and remains less understood (Sassano et al, 2017). Several proteins have been proposed to be involved in the tethering and stabilization of the contact sites. ER-resident Mfn2, for instance, was reported to tether the organelles by homo- and heterotypic interactions with mitochondrial Mfn1 and Mfn2 located at mitochondria (de Brito & Scorrano, 2008). The $Ca^{2+}$ receptor IP3R in the ER membrane is physically linked to VDAC1 in the mitochondrial outer membrane (MOM) by Grp75 (Szabadkai et al, 2006), and this interaction seems to be crucial for the efficient uptake of ER-released $Ca^{2+}$ into mitochondria. Recently, a new protein termed PDZD8 was identified as an MERC core component involved in tethering between the two organelles (Hirabayashi et al, 2017). Besides determining the components that act as tethers, other features of MERCs such as their dynamic spatiotemporal regulation, heterogeneity in composition and function, and their role in disease are yet to be established.

[1]Interfaculty Institute of Biochemistry, University of Tübingen, Tübingen, Germany   [2]University of Heidelberg, Heidelberg, Germany   [3]Center for Plant Molecular Biology, University of Tübingen, Tübingen, Germany   [4]Molecular Cognition Laboratory, Biophysics Institute, Consejo Superior de Investigaciones Científicas, University of the Basque Country (UPV)/Euskal Herriko University, Campus Universidad del País Vasco, Leioa, Spain   [5]Ikerbasque, Basque Foundation for Science, Bilbao, Spain   [6]Department of Physiology and Cell Biology and National Institute of Biotechnology in the Negev, Faculty of Health Sciences, Ben-Gurion University of the Negev, Beer-Sheva, Israel   [7]Max Planck Institute for Developmental Biology, Tübingen, Germany   [8]Hertie-Institute for Clinical Brain Research, University of Tübingen and German Centre for Neurodegenerative Diseases (DZNE), Tübingen, Germany

Correspondence: ana.garcia@uni-koeln.de
*Vanessa Hertlein and Hector Flores-Romero contributed equally to this work

Specific tools for membrane contact sites research are available and have contributed to our knowledge of MERCs. On the one hand, EM is one of the most accurate techniques to visualize membrane contact regions, but it is time-consuming, difficult to quantify, and only possible in fixed cells. Despite its wide applicability and possibility to use in living cells, visualization with confocal microscopy has the disadvantage of a resolution limit of around 200 nm, which makes data interpretation challenging (de Brito & Scorrano, 2008; Riccardo Filadi, 2015; Naon et al, 2016). Other methods such as proximity ligation assay are also limited to fixed cells and rely on the availability of high-quality specific antibodies (Gomez-Suaga et al, 2017). In yeast, Kornmann et al (2009) used the tethering complex ChiMERA with a GFP molecule flanked by a mitochondrial and an ER-targeting sequence to compensate for ERMES knockout. A next generation of MERC sensors is based on the fluorescence signal that increases only at the contact sites, by exploiting split (a split GFP-based contact site sensor [SPLICS]) or dimerization-dependent fluorescent proteins, or FRET coupled to MERC induction by rapamycin-dependent protein domain dimerization (FEMP) (Csordas et al, 2010; Alford et al, 2012; Toulmay & Prinz, 2012; Eisenberg-Bord et al, 2016; Cieri et al, 2018; Yang et al, 2018). However, these methods also have drawbacks, most importantly because the establishment of artificial physical links between the ER and the mitochondrial membrane can affect the composition, dynamics, stability, and regulation of the MERCs under investigation, thereby leading to artifacts. In addition, the establishment of this physical link between the two organelles is in many cases irreversible and limits their application to study MERC dynamics. Although the FRET-based probe FEMP theoretically would not be limited by these disadvantages, it seems that in practice, the low signal-to-noise ratio limits the calculation of reliable absolute FRET values, and the induction of artificial links via the autophagy inducer rapamycin is used to set maximum reference FRET values, which limits its application in living systems.

Here, we present a novel bioluminescence resonance energy transfer (BRET)–based biosensor for the analysis of distances between the mitochondria and ER, and therefore, for probing MERCs, which we call MERLIN (Mitochondria–ER Length Indicator Nanosensor). BRET is a variant of the well-established FRET technique that follows the same physical principle of the radiation-free energy transfer between two chromophores with overlapping spectra in close proximity (less than 10 nm). In BRET, however, the donor is the enzyme luciferase which oxidizes a substrate, the bioluminophore (Pfleger & Eidne, 2006), which then is able to transfer the energy to the acceptor by resonance. The donor and acceptor emission are then detected and quantified as the ratio of acceptor to donor emission. This ratio provides an estimation of the effectiveness of the transfer of the donor energy to the acceptor and thereby of the distance between them. Unlike with FRET, BRET biosensors do not require sample illumination to excite the donor, which reduces phototoxicity and cross talk with the excitation and detection of donor and acceptor. BRET is also independent of the orientation between donor and acceptor. These factors impact the efficiency of energy transfer and increase the signal-to-noise ratio. During the last decades, BRET has emerged as a powerful tool for the study of protein–protein interactions in vitro and in different physiologically relevant scenarios (Perroy et al, 2004; Coulon et al, 2008).

The main advantage of MERLIN, compared with other methods, is that it generates a BRET signal with a signal-to-noise ratio that is sufficient to enable sensing the proximity between the mitochondria and the ER without forcing interaction or establishing artificial connections at the MERCs. Because of this, MERLIN can be used to follow dynamics and reversibility of MERC formation and dissociation, which also sets it apart from other approaches. The two parts of the BRET biosensor are anchored to either the mitochondrial or ER membranes and each contain a protein of the BRET pair, Renilla Luciferase 8 (RLuc), or mVenus. A fully synthetic linker system with lengths between 0- and 24-nm spans the distance between the two organelles. To validate the functionality of MERLIN, we confirmed that MERC disruption by knockdown of PDZD8 was sensed by a decrease in the BRET signal. In addition, the biosensor detected an increase in the proximity of the ER and mitochondria when PDZD8 was overexpressed, when MERCs were forced by expression of a synthetic linker as well as during apoptosis. We demonstrated the applicability of MERLIN to detect dynamic changes in the distance between the mitochondria and ER by quantifying the reversible responses to a number of cellular stresses. We also report the applicability of MERLIN in sensitive cell types such as living neuronal progenitors and neurons. Finally, we used MERLIN to investigate the role of the machinery for mitochondrial dynamics in MERCs. We found that knockdown of mitofusins 1 and 2 (Mfn1 and Mfn2) or dynamin-related protein 1 (Dp1) resulted in a decrease of the BRET signal, underscoring the importance of mitochondrial shape and dynamics for the maintenance of the contact sites. Altogether, MERLIN is a powerful and innovative tool for the investigation of the mitochondria–ER membrane contact sites.

## Results

### Rational design and systematic optimization of BRET-based sensors of proximity between the ER and mitochondrial membranes

To develop a new tool that allows studying the distance between mitochondria and the ER membrane with minimal interference, and therefore, also their contact sites, we developed a BRET-based biosensor with RLuc acting as a donor and mVenus as an acceptor. We generated MERLIN, a modular, genetically encoded system, where each of the two components of the BRET pair was targeted to the MOM or to the ER membrane. MOM targeting was achieved via the C-terminal domain of the Bcl-2 family protein Bcl-xL (further termed B33C, Bcl-xL C-terminus 33aa) (Kaufmann et al, 2003). For the ER localization, we used a truncated nonfunctional variant of calnexin (termed hereafter as sCal), an ER chaperone, which consists of the ER-targeting sequence and the cytosolic C terminus but lacks most of the ER-luminal N-terminus. In order to bridge the distance between the two organelles at the contact sites we used a fully synthetic linker system with different lengths (0–12 nm). The linker consists of amino acid repeats with the sequence A(EAAAK)nA and forms a α-helical structure that is laterally stabilized by salt bridges between the glutamate and lysine residues (Marqusee & Baldwin, 1987; Kolossov et al, 2008). Three different variants of the linker were designed as L1 with a theoretical length of 3 nm and L2

and L3 with a theoretical length of 6 nm. Using different combinations of this linker system, a distance of up to 24 nm plus the length corresponding to the size of RLuc and mVenus and the connection between the membrane anchors and the linker system can be spanned (Fig 1A), which should be sufficient to cover the separation between the mitochondria and ER membranes at MERCs (Csordas et al, 2006).

To confirm the correct intracellular targeting of the MERLIN components, all constructs of the biosensor system were expressed in Cos1 cells and visualized by confocal microscopy. As expected, the mVenus and the RLuc constructs, immunostained with an anti-RLuc antibody, co-localized with MitoTracker and GRP78, respectively, indicative of mitochondrial or ER distribution according to their targeting signal (Fig 1B and C).

Next, we characterized the effect of the biosensor expression on cell viability by analyzing the release of the apoptotic protein Smac tagged with mCherry, under healthy and apoptotic conditions. Consistent with the nature of the BRET-based MERLIN partners, the overexpression of these constructs did not affect cell viability neither in healthy nor under apoptotic conditions (Fig S1).

After verifying their correct localization and negligible effect on cell viability, we carried out a systematic analysis of the biosensor performance using quantitative saturation BRET assays. For these experiments, we used cells co-expressing constant amounts of the donor protein and increasing amounts of the acceptor protein. We calculated the BRET ratio as the acceptor emission relative to the donor emission and corrected by subtracting the background ratio value detected when only RLuc was expressed.

To find out the optimal pair of biosensor components that is most sensitive despite the heterogeneity in ER/mitochondria distances, we performed BRET saturation assays for all possible linker combinations. We quantified the BRET signal for BRET pairs coupled to 0-, 6-, 12-, and 24-nm linkers, as well as with donor/acceptor targeted to the ER/mitochondria and vice versa. The quantitative BRET assays showed a saturation curve for all linker lengths, indicating specificity (Figs 2A and B, and S2). We detected the strongest BRET signal for the biosensor pairs based on 6- and on 12-nm linker length. Interestingly, the BRET ratios were about three times higher for all linker lengths when the donor was localized to the ER (Fig 2C). This difference might be due to the active co-translational insertion of ER membrane proteins compared with the passive post-translational insertion of MOM proteins to different expression levels of donor and acceptor in the two organelles or to a potential effect of redox nanodomains (Booth et al, 2016) on the luciferase reaction. To control that indeed the ROS levels do not affect MERLIN activity, we compared the luciferase activity in cells stably expressing MERLIN under normal and hypoxic conditions and confirmed that the signal was not significantly changed (Fig S3).

As a negative control, we measured BRET saturation curves for biosensor combinations in which the donor and acceptor fragments were spatially separated by targeting them to two different cellular compartments. As expected, cells co-expressing the donor in the ER (sCal-L1-RLuc) and the acceptor either facing the lumen of the ER (mVen-ER5) or localized to the nucleus (mVen-H2B6) showed extremely low BRET ratios (Fig 2D).

As a positive control for maximum BRET, we prepared constructs in which the donor and acceptor proteins where physically linked,

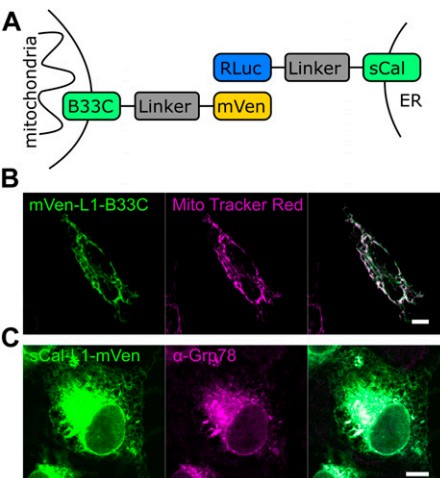

**Figure 1. Rational design of the MERLIN system and subcellular localization of its components.**
**(A)** Scheme illustrating the structure of the BRET biosensors. The mitochondrial part of the biosensor is targeted to the MOM by the alpha-helical C-terminal domain of Bcl-xL (B33C). For ER targeting, a truncated nonfunctional variant of calnexin (sCal) is used. A fully synthetic linker system which can be combined in different ways to span a distance of up to 24-nm connects the membrane domain to the proteins of the BRET pair. **(B)** Confocal image of an individual Cos1 cell expressing the mitochondrial biosensor mVen-L1-B33C (green). Mitochondria were stained with MitoTracker Red (magenta). **(C)** Confocal image of an individual Cos1 cell expressing the ER biosensor sCal-L1-mVen (green). ER was immunostained with anti-Grp78 antibody (magenta). Scale bar 10 μM.

which was achieved by expressing them as a single polypeptide (Rluc-L1-mVen). As expected, cells expressing the construct RLuc-L1-mVen showed much higher BRET signal than all other biosensor combinations tested at the same donor/acceptor ratio (Fig 2E). Of note, the BRET signal of the positive control in Fig 2E is lower than the maximum BRET ratio of MERLIN in Fig 2C, but this is due to the equimolar ratio of the donor and acceptor in the fusion-construct RLuc-L1-mVen (the highest BRET ratios were obtained at a donor/acceptor ratio of 1:6, Fig 2B).

## Validation of MERLIN

To demonstrate the applicability of MERLIN to study mitochondria/ER contact sites, it is important to validate that the sensor responds with significant signal changes under cellular settings that are known to affect MERCs. For this purpose, we analyzed the sensitivity of MERLIN to changes in the levels of PDZD8, a known tether of MERCs (Hirabayashi et al, 2017), and to induction of MERCs with a synthetic linker. As expected, considering its ability to tighten ER–mitochondria membranes, the overexpression of PDZD8 significantly increased the BRET signal, whereas knocking PDZD8 down decreased it (Fig 3A–D). Furthermore, the overexpression of the synthetic tether mTagBFP2, which physically links the ER and mitochondrial membranes and robustly promotes the contacts between them (Hirabayashi et al, 2017), also enhanced BRET signal to a similar extent than PDZD8 overexpression (Fig 3D). The expression of the acceptor is increased linearly in a concentration-dependent manner, and it is not affected by the overexpression of the synthetic tether mTagBFP2 (Fig S3A and B).

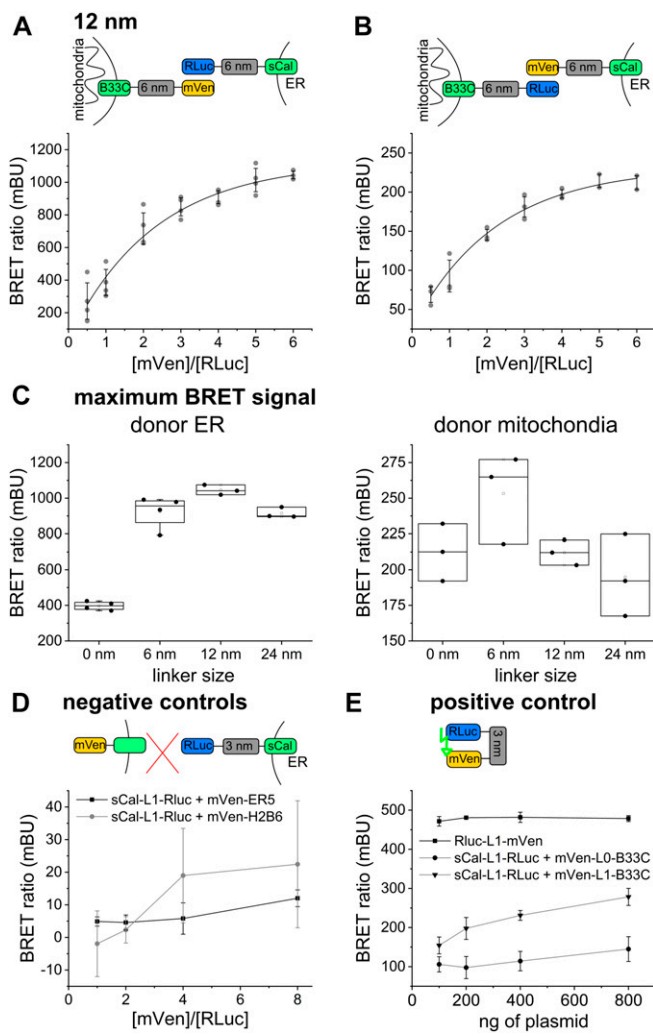

**Figure 2. Systematic optimization of MERLIN.**
**(A, B)** Scheme and saturation curve for MERLIN based on the 12-nm linker with (A) the donor targeted to the ER and the acceptor targeted to mitochondria and (B) the donor targeted to mitochondria and the acceptor targeted to the ER. **(C)** Maximum BRET signals for the different linker lengths and organelle localizations of the MERLIN components. **(D)** BRET signal for the negative controls sCal-L1-RLuc (3-nm donor) and mVen-ER5 (luminal ER protein) or mVen-H2B6 (nucleus). **(E)** BRET signal of the positive control mVen-L1-RLuc compared with the 3- and 6-nm linker lengths.

Furthermore, previous studies have shown that the contacts between mitochondria and the ER increase under apoptotic conditions (Csordas et al, 2006). To check if MERLIN could detect these changes, we first examined the kinetics of the process in Cos1 cells undergoing apoptosis upon staurosporine (STS) treatment by imaging over time. Under our experimental conditions, we observed dramatic fragmentation of the mitochondrial network about 1 h after cell death induction and cell body shrinkage after 5 h (Fig 3E). In agreement with this temporal evolution, we co-transfected the same amount of the donor and acceptor plasmids of the MERLIN system, induced apoptosis with STS, and monitored the BRET signal for up to 5 h. We observed an increase in the BRET signal of the apoptotic cells over time, whereas no significant changes were detected in control cells without apoptosis induction (Fig 3F). This

behavior was reproducible when using MERLIN combinations with 0-, 6-, 12-, and 24-nm linker lengths (Fig S4). To control that the increase in BRET is not due to cell shrinkage during apoptosis, we used QVD a pan-caspase inhibitor that blocks cell contraction upon STS treatment and confirmed a comparable increase in BRET (Fig 3D).

To validate MERLIN using conditions that are known to reduce MERCs, we treated cells with N-acetylcysteine (NAC), a compound that improves mitochondrial function and is accompanied by a decrease in contact sites between the ER and mitochondria. Accordingly, we could detect a significant amount in the BRET signal that was concentration dependent (Fig 3G).

Finally, we validated MERLIN with an alternative method using EM (Fig S3C). We first confirmed that expression of MERLIN did not alter the MERCs compared with wild-type cells. Then, we incubated the cells with tunicamycin or under starvation conditions, two treatments known to increase MERCs (Csordas et al, 2006; Yang et al, 2018). In these experiments, we could detect an increase in the BRET signal with MERLIN (Fig 4A), which was indicative of a tightening between the ER and mitochondria membranes, as confirmed by the increase in the ratio between MERCs and mitochondria quantified by EM (Fig S3C).

Altogether, these experiments confirmed that the MERLIN system is indeed able to detect a tightening or loosening of the contact sites between the mitochondria and ER under a number of perturbations that are known to affect MERCs and demonstrate the validity of the new biosensor.

### Characterization of MERC dynamics via MERLIN and use of MERLIN in sensitive cell types

The absence of a physical link between the two components of MERLIN should allow the biosensor to detect dynamic changes in the distance between the ER. To test if this is the case, we treated the cells transiently with several stimuli that have been proposed to modulate MERC formation and disruption and measured the BRET signal over time. For this purpose, we created a MERLIN-containing stable cell line, which exhibits correct organellar distribution and an insignificant effect in cell viability (Fig S3D–F). In coherence with previous results (Csordas et al, 2006; Yang et al, 2018), both tunicamycin treatment and starvation increased the BRET signal and the elimination of tunicamycin or starvation conditions reconstituted normal ER–mitochondria distances after 16 h, according to the return of the BRET signal to pretreatment values (Fig 4A, light blue and green lines). Bortezomib, also known as PS-341, is a proteasome inhibitor that induces unfolded protein response and ER stress (Teicher et al, 1999). In our system, bortezomib treatment induced a sharp decrease in the BRET signal after 4 h, which was restored upon stimulus removal (Fig 4A, purple line). Interestingly, bortezomib and tunicamycin induce ER stress by different mechanisms, which could be the reason why they induce opposite effects in the MERLIN signal. Bortezomib is a potent inhibitor of the 26S proteasome that induces ER stress as a secondary effect, whereas tunicamycin inhibits N-linked glycosylation and thereby blocks protein folding and transit through the ER. Moreover, the addition of Taxol, a potent cytoskeletal drug used in chemotherapy, significantly altered the BRET signal, which points out a direct link between cytoskeleton and MERC dynamics (Fig 4A, orange line).

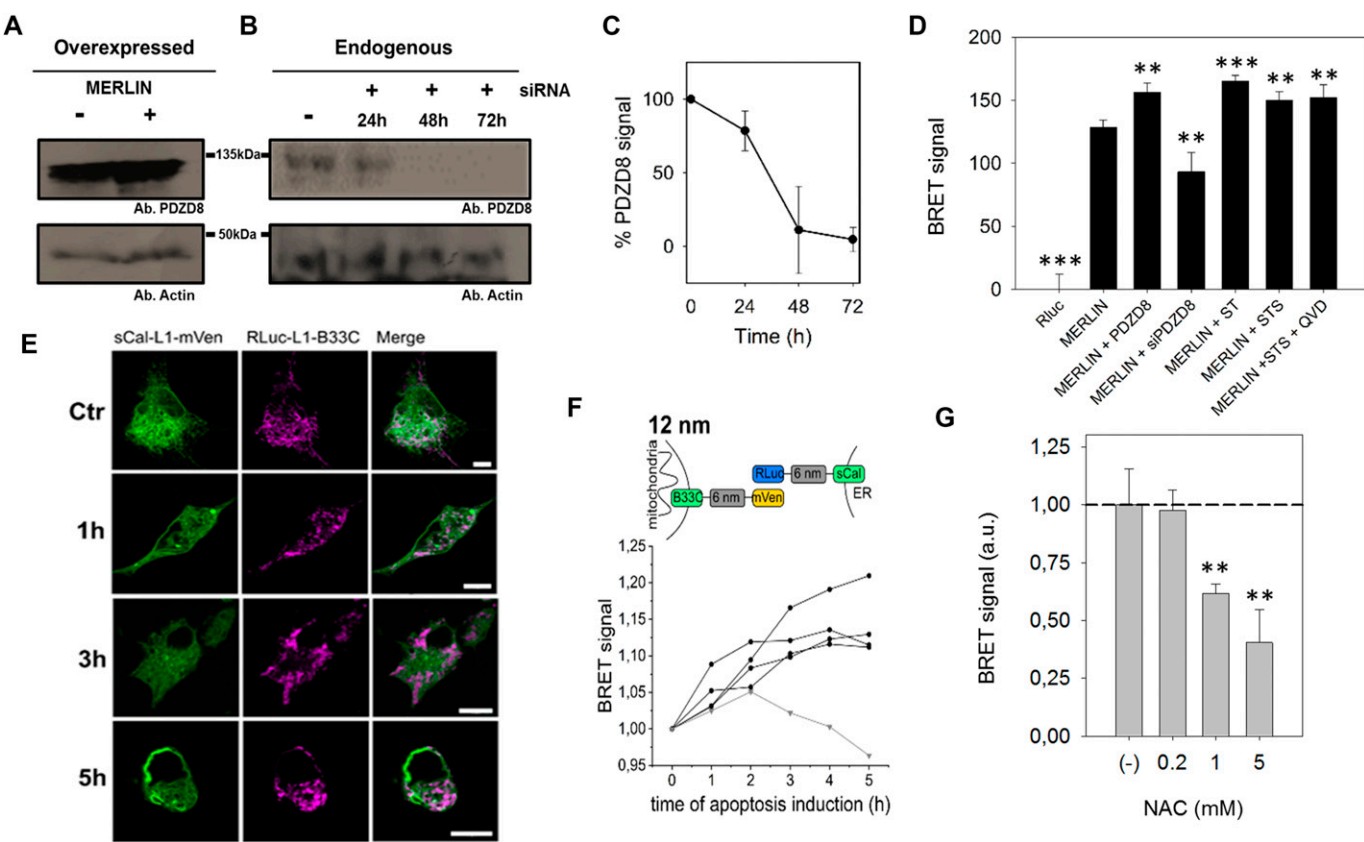

**Figure 3. Validation of MERLIN.**
**(A, B, C, D)** PDZD8 modulates ER–mitochondria distance. **(A, B, C)** Representative Western blot of the PDZD8 levels when transiently transfected and (B) upon silencing with siRNA_PDZD8 in HCT116 cells, whose quantification is shown in (C) (n = 3). **(D)** BRET signal in cells co-expressing Rluc-L1-B33C and Scal-L1-mVenus biosensor combination, in the presence of overexpressed PDZD8, the synthetic tether mTagBFP2 and PDZD8 knockdown in HCT116 cells. (**P < 0.025, ***P ≤ 0.001). *t* test, data are expressed as mean ± SD. **(E, F)** The BRET signal of MERLIN is increased in apoptotic cells. **(E)** Confocal images of Cos1 cells transfected with sCal-L1-mVen (green) and RLuc-L1-B33C (magenta) under healthy condition and upon apoptosis induction with 1 µM STS at different times. Scale bar 10 µM. **(F)** Scheme and graph showing the change of the BRET signal in apoptotic cells over time for the 12-nm linker MERLIN. Black lines represent four individual measurements and the grey line the control measurement without induction of apoptosis. Apoptosis was induced at time point 0 h by addition of 1 µM STS. (N = 4). **(G)** MERLIN detects a NAC-induced decrease in MERCs (**P < 0.025) *t* test, data are expressed as mean ± SD.

Interestingly, our data show that under all conditions tested, stimulus deprivation restored the BRET signal, which supports the high plasticity of MERCs and suitability of MERLIN to study MERC dynamics. In contrast, hypoxia did not affect the BRET signal (Fig 4A, white dots), suggesting that ROS levels do not affect MERCs (neither

the Luciferase activity nor the BRET signal). The increase in BRET signal upon STS treatment could not be recovered in agreement with the irreversibility of apoptosis (Fig 4A, blue dots). As control, we confirmed that the treatments alone did not affect significantly the RLuc activity at the concentrations and conditions tested (Fig S3G)

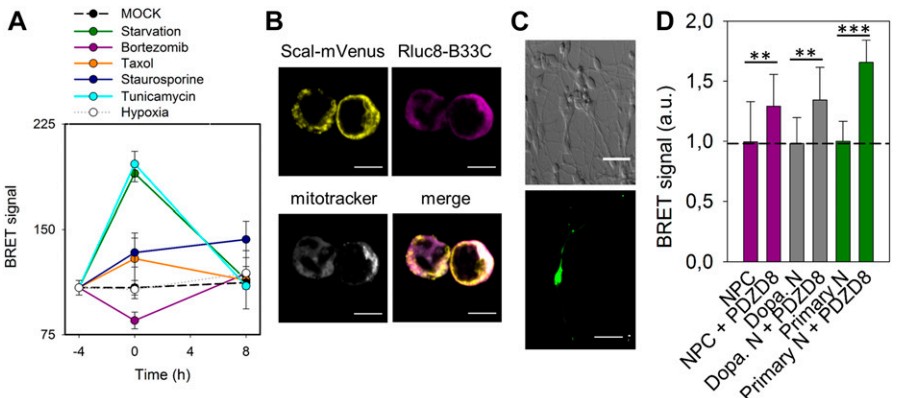

**Figure 4. MERC plasticity characterized by MERLIN in stable HCT116 cells and use of MERLIN in neuroprogenitors and dopaminergic neurones.**
**(A)** Measurement of BRET signal of MERLIN as a function of time in HCT116 cells exposed to stress: starvation (green), bortezomib (purple), Taxol (orange), staurosporine (dark blue), tunicamycin (cyan), and hypoxia (grey). Control shown in black. BRET was quantified before treatment (−4 h), after 4 h of stress (0 h) and upon recovery at 4 and 16 h. **(B)** Localization of the donor and acceptor to the mitochondria and ER, respectively in neuroprogenitor cells. Scale bar 5 µm. **(C)** Representative image of a differentiated dopaminergic (top) and embryonic mice primary neurons (bottom). Scale bar 100 and 20 µm, respectively. **(D)** Quantification of BRET signal in neuroprogenitor cell (magenta) and dopaminergic neurons (grey) in the presence of absence of PDZD8. (**P < 0.025 and ***P < 0.001). *T* test, data are expressed as mean ± SD.

and that the coelenterazine H added did not significantly affect the BRET signal (Fig S3H).

Studying MERCs in sensitive cell type such as neurons remains challenging because of the problems with phototoxicity in FRET-based biosensors and the difficulties to apply EM. These issues can be overcome by MERLIN, which we used to detect changes in MERCs induced by PDZD8 overexpression in neuronal progenitors and in differentiated neurons (Fig 4B–D). Altogether, these experiments support the wide applicability of MERLIN.

## MERLIN design is compatible with FLIM-FRET analysis of ER–mitochondrial distance in single cells

BRET saturation assays are a perfect technique for high-throughput screenings in multi-well plate formats. However, we also wanted to test if MERLIN was compatible with light microscopy and the quantification of membrane contact sites in single cells (Fig 5A). Because it is not trivial to detect bioluminescence with light microscopy, we exchanged the BRET pair for a FRET pair (mCerulaen3 and mVenus) in the modular biosensor system.

We measured the proximity between the ER and mitochondria in experiments of fluorescence lifetime imaging (FLIM)-FRET using cells co-expressing biosensor combinations based on the 6-nm linker and with the donor targeted to the ER or to the mitochondria. We compared the fluorescence lifetime of the donor in these cells with that of cells only expressing the donor or the acceptor as negative controls (mCer-L1-B33C or sCal-L1-mVen). As additional positive and negative controls, we measured the donor fluorescence lifetime in cells expressing a donor–acceptor construct (mCer-L0-mVen) and in cells co-expressing spatially separated donor and acceptor (mCer-L1-B33C + A2A-mVen). As shown in Fig 5B, the fluorescence lifetime of the donor in cells expressing spatially separated biosensor fragments (3.70 ± 0.06 ns) was comparable with that of cells expressing donor only (3.67 ± 0.08 ns) and was slightly lower than reported lifetime values in absence of FRET (Markwardt et al, 2011). In contrast, the fluorescence lifetime of the donor in the MERLIN system was significantly shorter than the lifetime of the two controls (3.50 ± 0.05 ns for mCer-L1-B33C + sCal-L1-mVen and 3.54 ± 0.06 ns for sCal-L1-mCer + mVen-L1-B33C), which indicates FRET between the two sensor components resulting from the juxtaposition of the ER and mitochondria. The donor–acceptor construct, mCer-L0-mVen, showed the most efficient non-radiant energy transfer and, thus, the shortest fluorescence lifetime of the donor (3.06 ± 0.17 ns). These results show that also in single cells, the FLIM-FRET–based MERLIN allows the quantitative analysis of the proximity between the mitochondria and ER.

## Role of the machinery for mitochondrial dynamics on MERC regulation

The mitochondria–ER interface contains proteins involved not only in the tethering and regulation of MERCs but also proteins responsible for the several biological functions performed at these sites. Although the molecular composition remains enigmatic, the machinery for mitochondrial dynamics has been associated with MERCs (de Brito & Scorrano, 2008; Friedman et al,

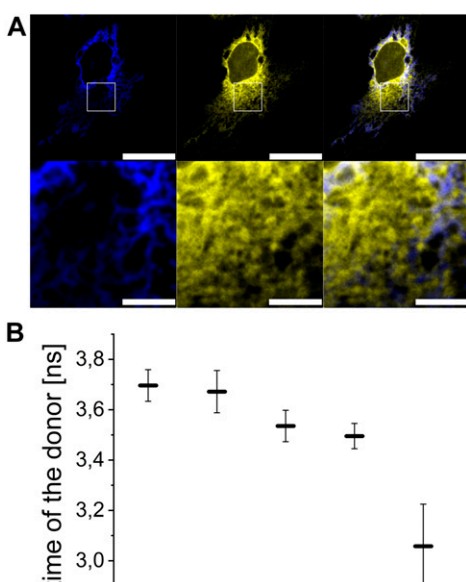

**Figure 5. FLIM-FRET of MERLIN.**
**(A)** Upper plane shows a representative Cos1 cell transfected with mCer-L1-B33C (blue) and sCal-L1-mVen (yellow). Scale bar 25 μM. The area in the white rectangle was used for FLIM-FRET measurement. Lower plane shows the zoom in this area. Scale bar 5 μM. **(B)** The fluorescence lifetime is shown for the donor fluorophore with the 6-nm linker MERLIN, the negative and the positive control as well as the donor only control. Graph shows three biological replicates with n = 10, Error bars SD.

2011; Elgass et al, 2015; Riccardo Filadi, 2015; Naon et al, 2016). Several lines of evidence connect the proteins responsible for MOM fusion, Mfn1 and Mfn2, with membrane tethering at MERCs. However, their role in the tethering is debated and two opposite models have been proposed. In one hypothesis, both Mfn1 and Mfn2 act as heterotypic ER/mitochondria tethers at contact sites, whereas in the second model, these proteins rather behave as antagonists of a tether (de Brito & Scorrano, 2008; Riccardo Filadi, 2015; Naon et al, 2016).

To shed light on this issue, we compared the BRET signal of cells expressing MERLIN and knocked down for Mfn1 or Mfn2 with that of control cells without knockdown or with scramble siRNA knockdown as negative control (Figs 6 and S5). Mfn2 is located at both the ER and mitochondrial membranes, whereas Mfn1 localizes exclusively to the MOM (de Brito & Scorrano, 2008). Interestingly, we found that cells with Mfn1 or Mfn2 siRNA knockdown showed fragmented mitochondria and slightly altered ER morphology compared with the control cells (Fig 6A) without affecting the localization of the RLuc (Fig S6). Furthermore, we measured a lower BRET ratio for both Mfn1 and Mfn2 siRNA knockdown cells compared with control cells (Fig 6B). These results indicate a decrease in the proximity between the ER and mitochondria in cells with reduced levels of Mfn1 or Mfn2 and, therefore, support a role of Mfn1 and Mfn2 in promoting MERCs. From these experiments, however, we cannot exclude the possibility that the changes in the BRET signal indirectly

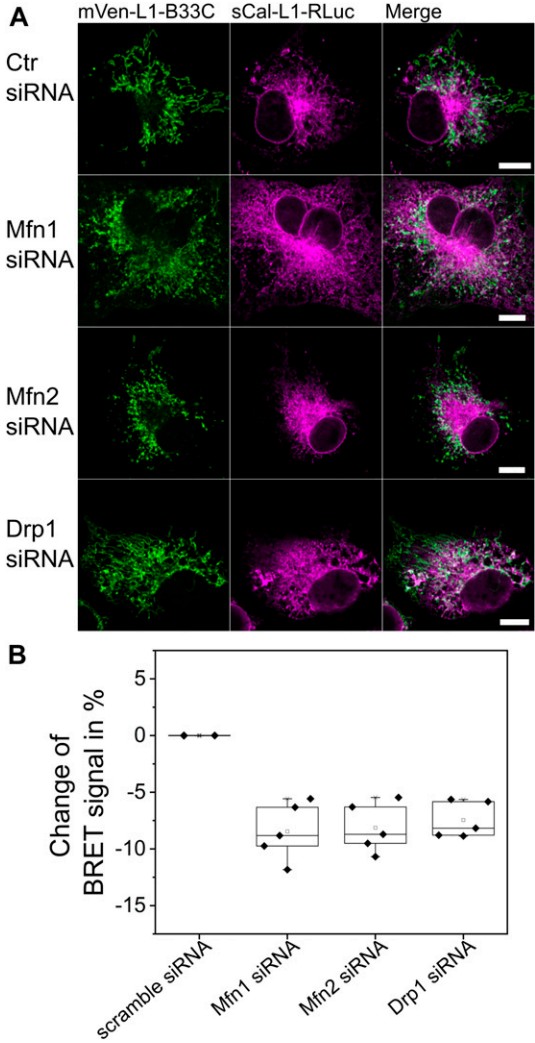

**Figure 6. siRNA knockdown of proteins involved in mitochondrial dynamics alters the BRET signal of MERLIN.**
**(A)** Confocal images of Cos1 cells after Mfn1, Mfn2, Drp1 knockdown, or scramble (Ctr) siRNA transfection. Scale bar 10 μM. **(B)** Changes in percentage of BRET signal in cells co-expressing the 12-nm linker MERLIN sCal-L2-RLuc and mVen-L2-B33C after knockdown with Mfn1, Mfn2, Drp1, or scramble (Ctr) siRNA normalized to cells without knockdown. n = 3–4, Error bars SD.

result from the alterations in the morphology of the mitochondrial network that has an effect on the contacts with the ER.

In addition to Mfn1 and Mfn2 knockdown, we also tested the effect of siRNA knockdown of the mitochondrial fission protein Drp1 on the proximity between the ER and mitochondria measured with MERLIN. Drp1 has been reported to be recruited at MERCs to mediate mitochondrial division (Friedman et al, 2011), but a potential additional role in MERC regulation remains unclear. As expected, knockdown of Drp1 produced elongated mitochondria (Fig 6A) without affecting the localization of the RLuc (Fig S6). However, this also resulted in decreased BRET signal compared with control cells (Fig 6B).

Altogether, our findings demonstrate the applicability of the MERLIN system to study the association between the ER and mitochondria. Using the biosensor, we show that the machinery involved in mitochondrial fusion and fission affect the contacts

between both organelles, which is associated not only to a likely tethering role of mitofusin 2 but also to alterations in the morphology of the mitochondrial network regulated by these proteins.

## Discussion

Here, we present MERLIN, a novel modular biosensor system for probing the proximity between the ER and mitochondria, which is based on BRET between RLuc and mVenus targeted to each of the organelle membranes in a complementary manner. The BRET signal depends on the distance between donor and acceptor, which should be within a radius of at most 10 nm for efficient energy transfer. In MERLIN, they are brought together by a modular linker system that can be tuned to span different lengths ranging from 0 to 24 nm, plus the size of the donor/acceptor proteins and that of the membrane anchors. Although the linkers in MERLIN are designed to structure into rigid rods (Marqusee & Baldwin, 1987; Arai et al, 2001), the short connecting regions to the membrane anchors and to the donor/acceptor are flexible and allow rotation on the membrane plane and bending. As a result, the MERLIN modular system can adopt a distribution of 3D conformations that enable BRET over a range of distances between the ER and mitochondria below a threshold set by the sensor components in their most extended conformation. These considerations may not have been taken into account in the design of other proximity sensors between the ER and mitochondria.

We validated the sensitivity of MERLIN to probe changes in the distance between the ER and mitochondria, and thereby sense contact sites, by inducing a number of cellular perturbations that are known to promote concrete alterations in MERCs. We confirmed that overexpression or knockdown of PDZD8, a recently discovered mitochondria/ER tether and core component of MERCs (Hirabayashi et al, 2017), increased or decreased the MERLIN signal, respectively. The biosensor also detected the increase in proximity between the two organelles that has been reported to occur during apoptosis (Csordas et al, 2006). Finally, the promotion of MERCs via a synthetic linker (Hirabayashi et al, 2017) resulted in an increase of the BRET signal too. These validation experiments prove the sensitivity of MERLIN to changes in the distance between the ER and mitochondria under different cellular settings. Furthermore, we successfully validated the results obtained with MERLIN with an alternative method by quantifying the contact sites from EM images.

The most important feature of MERLIN that sets it apart from alternative biosensors currently available (Csordas et al, 2010; Alford et al, 2012; Cieri et al, 2018; Yang et al, 2018) is that it does not depend on the formation of a physical connection that bridges the ER and mitochondria. This avoids potential unwanted effects induced by the enforced linkage, which could alter MERC composition, dynamics, and/or regulation, or even affect the cellular homeostasis (Pinton et al, 2008; Grimm, 2012).

A second advantage of MERLIN over other systems, precisely related to the absence of a physical connection between the two sensor components, is that it allows studying reversible processes. This is the formation and dissociation of MERCs and the regulation of their dynamics. Here, we demonstrated the ability of MERLIN to follow the plasticity of MERCs by following the kinetics of BRET

changes resulting from transiently treating the cells with stress inducers over time. Our results indicate that cells are able to recover a steady state in the distance between the ER and mitochondria once the stress stimulus is removed.

It is important to note that MERLIN is a sensor of proximity, and it is not specific to contact sites. ER and mitochondria that are proximal to each other without any tether will also produce BRET. Nevertheless, because MERCs are characterized by a short separation between the two organelles, they are expected to be the major contributors to the BRET signal. Indeed, in our validation experiments, we demonstrate that MERLIN is a sensitive system capable of probing changes in MERCs. In this sense, the MERLIN system provides information related to the total juxtaposed area between the mitochondria and ER, but not about the number, size, or dynamics of individual contact sites. Along the same lines, the BRET sensor is not specific for different type of MERCs and cannot differentiate if the contacts have distinct molecular compositions.

Using BRET as output signal has the advantage that no donor illumination is required, which avoids problems of phototoxicity and cross talk with the acceptor excitation and emission. The BRET signal is robust and, unlike with FEMP, no addition of rapamycin to maximize the signal by artificial mitochondria/ER juxtaposition is needed (Csordas et al, 2010). As a result, it also includes measurements of living cells, including sensitive cell types such as neurons shown here, at different time points during biological processes and even kinetic measurements if the adequate RLuc substrate is used (Pfleger & Eidne, 2006). Furthermore, we demonstrate here how MERLIN is especially convenient for measurements in multi-well plates, which simplifies high-throughput genetic and drug screenings. The combination of MERLIN with microscopy could be of interest in some instances, for example, when the study of contact sites is to be combined with organelle morphology analysis at the single cell level. Although it is difficult to visualize BRET in microscopic studies because of low levels of light emission and a lack of sensitivity of many cameras, MERLIN can be adapted to imaging strategies by exchanging RLuc for mCerulean and thereby transforming the system in a FRET sensor, although the signal-to-noise ratio is lower. Here, we show how MERLIN is also sensitive to MERCs by FLIM-FRET. Other forms of FRET that do not require special instrumentation, such as acceptor photobleaching or sensitized emission FRET could be possible too.

Mitochondrial morphology and the machinery regulating have been reported to affect MERCs (Lee & Yoon, 2014). Mfn2, which is part of this machinery by mediating mitochondrial fusion, has also been proposed to act as a tether between the ER and mitochondria (de Brito & Scorrano, 2008; Naon et al, 2016). Alternative studies suggest that it rather acts as an antagonist of MERCs, but the debate remains unsettled (Riccardo Filadi, 2015; Leal et al, 2016). Here, we used MERLIN to understand how Mfn2 and other proteins responsible for mitochondria fusion and fission affect the proximity between this organelle and the ER. If one reasons that the main effect of Mfn2 on MERCs is its role as a tether, one would expect that Mfn1 knockdown, which still allows for heterotypic ER/mitochondrial association via Mfn2 located at both organelles, would have a relatively lower effect on the average distance between them. However, we found that both Mfn2 and Mfn1 knockdown led to mitochondrial

fragmentation and to a similar decrease in the BRET signal. In contrast, Drp1 knockdown promoted elongated mitochondria, yet it also decreased the BRET signal, which brings the question whether any alteration in mitochondrial dynamics or shape strongly affects the contacts with the ER. Altogether, these results suggest that despite Mfn2 acting or not as a tether, the mitochondrial alterations induced by its deletion or overexpression have a dominating effect on MERCs and the overall distance between the ER and mitochondria.

To conclude, here, we present MERLIN, a novel proximity sensor for the distances between the ER and mitochondria, which is sensitive to alterations induced by genetic or pharmacological treatments. The main advantages of MERLIN compared with current alternatives are that it does not rely on any physical connection between the two organelles and that it can be used to study reversibility of MERCs. This modular biosensor approach could be easily extended to probe other inter-organelle contact sites by exchanging the targeting signals of the complementary components and selecting the optimal linker length. MERLIN opens the possibility to implemented inter-organelle proximity sensors in in vivo models such as mice because bioluminescence detection has been well established in these systems. Finally, we demonstrate the applicability of MERLIN by examining the role of the machinery for mitochondrial dynamics on the juxtaposition between the ER and mitochondria.

# Materials and Methods

## Antibodies

Commercial antibodies used in this study were anti-Grp78 (Abcam), anti-RLuc (Abcam), anti-Mfn1 (Cell Signaling Technology), anti-Mfn2 (Cell Signaling Technology), anti-Drp1 (BD Bioscience), anti-PDZD8 (PA5-46771; Thermo Fisher Scientific), and anti-$\beta$-actin (A2228; Sigma-Aldrich).

## Construction of plasmids

pcDNA3.1(-) (Invitrogen) served as general targeting vector for all constructs. TOPO-TA cloning was performed into the plasmid pCR2.1-TOPO (Invitrogen). Restriction enzymes NheI and BamHI were used for the insertion of the constructs into the pcDNA3.1(-) vector and restriction enzymes XbaI and EcoRI for the insertion of the linker sequence (Eurofins-MWG). All constructs for expression using Sindbis virus were synthetized in pSinRep5 (Thermo Fisher Scientific). Restriction enzymes Mlu1 and StuI were used for the insertion of the constructs into the SR5 vector. The plasmids mVen-ER-5 (#56611) and mVen-H2B-6 (#56615) and cDNA of PDZD8 (#105005) and mTagBFP2 (#105011) were purchased from Addgene. Smac-mCherry was a gift from Dr Stephen Tait (University of Glasgow) and the components of the BRET pair were a gift from Dr Peter McCormick (University of Surrey).

## Cell culture and transfection

Cos1, HCT116, and HCT116 cells containing MERLIN were maintained in DMEM (Invitrogen) and McCoy's5A (modified) medium (Sigma-Aldrich),

respectively, and supplemented with 10% FBS (Invitrogen) and 1% penicillin/streptomycin (Invitrogen). Cells were transfected with Lipofectamine 2000 (Thermo Fisher Scientific) at 60–80% confluence.

## Preparation of mouse primary neurons and neuron differentiation from human induced pluripotent stem cells

Primary neurons in culture were prepared from E18 Sprague Dawley rat hippocampi as described by Sanchez-Puelles et al (2019). Hippocampi were dissected and dissociated using trypsin (0.25%) and DNase I (0.1 mg/ml) and further subjected to mechanical trituration. Neurons were plated on 0.1 mg/ml poly-L-lysine–coated 24-well plates at a final density of $1.5 \times 10^5$ cells/well and 96-well plates at $6 \times 10^4$ cells/well. Neurons were maintained under 5% $CO_2$ at 37°C in Neurobasal medium (Gibco) supplemented with B27 (Gibco), FBS (Gibco), and GlutaMAX (Gibco) until 7 days in vitro (DIV), after which the medium was replaced with the Neurobasal medium supplemented only with B27. To avoid excessive glial proliferation, neurons were treated with the antimitotic cytosine arabinoside (5 $\mu$M; Sigma-Aldrich) after incubation for 7 DIV. Viral infection was performed in DIV21 neurons during 24–48 h.

Midbrain dopaminergic neurons were generated with a protocol adapted from Reinhardt et al (2013). IPSCs were cultured in 10 $\mu$M SB431542 (SB; Sigma-Aldrich), 1 $\mu$M dorsomorphin, 3 $\mu$M CHIR99021 (CHIR; Axon), and 0.5 $\mu$M purmorphamine (PMA; Alexis) on uncoated cell culture dishes to let them form embryoid bodies. Embryoid bodies were plated on Matrigel (Corning)-coated six-well plates in 150 $\mu$M ascorbic acid (AA; Sigma-Aldrich), 3 $\mu$M CHIR, and 0.5 $\mu$M PMA. After several passages, small molecule precursor cells (smNPCs) were obtained and cultivated in medium containing 150 $\mu$M AA and 3 $\mu$M CHIR99021. Differentiation of confluent smNPCs was initiated by cultivation in CHIR99021 free maintenance medium for 3 d, followed by 7 d in patterning medium containing 10 ng/ml FGF8 (Peprotech), 1 $\mu$M PMA, 200 $\mu$M AA, and 20 ng/ml BDNF (Peprotech). The differentiation was matured with BDNF, GDNF (Peprotech), TGFß-III (Peprotech), AA, dbcAMP (Applichem), and DAPT (Sigma-Aldrich). Before experiments, maturation medium was replaced 24 h before by N2 medium. All treatments were only performed in the N2 medium.

## Characterization of MERLIN subcellular localization and effect on cell viability by immunoblotting and confocal microscopy

Cos1 or HCT116 cells were grown on glass coverslips and transfected with MERLIN constructs for 16 h. For immunostaining, the cells were fixed at RT for 15 min with 4% paraformaldehyde and permeabilized by incubation with 0.25% Triton X-100 in PBS (PBST) for 10 min. If needed, before cell fixation, mitochondria were stained with 200 nM MitoTracker Red (Life Technologies) for 30 min at 37°C and 5% $CO_2$. Subsequently, the samples were blocked with 3% BSA in PBST (45 min at RT) and incubated with primary antibodies (1:100 in PBST with 3% BSA) for 1 h at RT. Next, the samples were washed with PBS, incubated with appropriate secondary antibody (1:200 in PBST) for 1 h at RT, and washed with PBST. In the cell viability experiments, the cells were grown as described above and transfected with Smac-mCherry and MERLIN (Smac/donor/acceptor in a 2:1:3 ratio). If required, the cells were treated with 1 $\mu$M staurosporine (STS) for 4 h at 37°C and 5% $CO_2$. In hypoxia experiments, redox was measured

upon BODIPY (Thermo Fisher Scientific, 1 $\mu$M) addition for 30 min at 37°C and 5% $CO_2$, in the presence/absence of 25 nM Monoethanolamin in HCT116 cells. Image acquisition was made with a Zeiss LSM 710 ConfoCor3 microscope (Carl Zeiss) equipped with a temperature and $CO_2$ controller using a C-Apochromat ×40 NA 1.2 water immersion objective (Zeiss) and Leica SP8 microscope with ×63 NA 1.5 oil immersion objective (Leica Microsystems GmbH). Excitation light came from argon ion (488 nm) or HeNe (561, 633 nm) lasers. Images were processed and analyzed with ImageJ.

## Generation of MERLIN-containing HCT116 stable cell line

HCT116 cells were transfected with Rluc-B33C and ScaI-mVenus for 16 h as described above and diluted up to individual colonies. Next, G418:McCoy's5A (modified) medium (Sigma-Aldrich) (0, 7 mg/ml) selection was carried out during 2–3 wk. Finally, we isolated single clones using the colony cylinders and checked for MERLIN presence and targeting by immunoblotting and by measuring the BRET signal.

## Sindbis virus purification

Sindbis virus was produced as described by Malinow et al (2010), with minor modifications. Briefly, BHK-21 cells were co-transfected with pSinRep5 RNA of interest and helper pDHtRNA. After 48 h, biosensor-containing viruses were collected and purified by a sucrose gradient. The samples were centrifuged for 90 min at 35,000 rpm (4°C) in an SW 60 Ti swinging-bucket rotor (Beckman Coulter) in a Beckman Optima L-100K. Viral particles were collected from 20%/55% sucrose.

## BRET measurements

In BRET assays, the cells were seeded in a white 96-well plate (#655073; Greiner) and transfected with MERLIN for 16 h or infected with MERLIN for 48 h. The cells were washed with PBS, incubated with 5 $\mu$M coelenterazine h (Promega) in PBS for 5 min in the dark and BRET measurements were carried out in a Tecan Infinite M200 plate reader at RT. If necessary, the cells were transfected with PDZD8:MERLIN or ST:MERLIN at equimolar concentrations. BRET signal was calculated as acceptor emission relative to donor emission and corrected by subtracting the background ratio value detected when RLuc is expressed alone. In the assay for characterization of MERC plasticity, HCT116 cells were transfected with MERLIN as described above. Next, the cells were treated with 15 $\mu$M Taxol, STS 1 $\mu$M, 50 nM bortezomib, 25 nM Monoethanolamin (hypoxia), 25 nM tunicamycin, or deprived of FBS (starvation) for 4 h at 37°C with 5% of $CO_2$. Then, BRET measurements were carried out and subsequently the media was removed and substituted by fresh media. The cells were then incubated for 4–16 h to allow for recovery and subsequently subjected to BRET analysis. NAC treatment was prolonged for 10 d by exchanging the media every 48 h.

## Transmission electron microscopy

The cells were seeded on Matrigel (Corning)-coated glass coverslips and cultivated for 2 d before transfection and drug treatment. After washing

and fixation with 2.5% glutaraldehyde (Sigma-Aldrich) in 20 mM Hepes buffer (pH 7.4) for 2 h at 37°C, the cells were washed with buffer, post-fixed in 2% osmium tetroxide, dehydrated, and embedded in epoxide resin (Araldite, Serva) as described previously (Wolburg-Buchholz et al, 2009). Ultrathin sections were performed using a Reichert Ultracut ul-tramicrotome (Leica) and were analyzed in an EM 10 electron micro-scope (Zeiss). Images were taken by a digital camera (Tröndle).

## Western blotting

Protein samples (50–200 μg protein) were separated by discon-tinuous 8.5–15% acrylamide SDS–PAGE and electrotransferred to a polyvinylidene fluoride membrane (no. ISEQ07850; Millipore) using a semi-dry Turbo-blot apparatus (Bio-Rad). The membrane was blocked at RT for 1 h and probed at 4°C overnight with the appropriate primary antibody. After washing with 1× TBST, the HRP-conjugated secondary antibody was added in 5% milk and incubated for 1 h at RT. The membrane was washed with 1× TBST and developed with ECL (Western Lightning Plus-ECL; PerkinElmer).

## Silencing assays

The cells were transfected with siRNA at a concentration of 2–10 nM for 48–72 h with Lipofectamine 2000 (Invitrogen) according to the manufacturer's recommendation. Scramble siRNA used as a control in silencing experiments was purchased by Dharmacon (D-001810-01-20). Specific siRNA for knocking down Mfn1 (J-010670-12-0002), Mfn2 (J-012961-05-0002), Drp1 has a customized sequence (GGAGCCAGCUAGAUAUUAAUU), and PDZD8 (L-018369-02-0005) were purchased from Dharmacon. After transfection, BRET measure-ments were carried out as described above. PDZD8 signal was quantified and normalized to the actin signal by ImageJ.

## FLIM-FRET

FLIM-FRET measurements were performed using a Leica TCS SP8 confocal microscope (Leica Microsystems GmbH) equipped with a FLIM unit (PicoQuant GmbH). For excitation (ex) and emission (em) of fluorescent proteins, the following laser settings were used: mCerulean3 at ex458 and em465–505 nm; mVenus at ex514 and em520–560 nm. FLIM data derive from three different biological replicates and measurements of 10 cells each replicate.

# Supplementary Information

# Acknowledgements

We thank Peter McCormick for helpful advice and discussion and Carolin Stegmüller, Sabine Schäfer, Iris Koch, Maria Zarani, Astrid Schauss, Christian Jüngst, Felix Babatz, and Marina Nikolova for technical support. This work has been partially supported by the Deutsche Forschungsgemeinschaft (FOR2036 GA1641/2-1 and GA1641/2-2) and the European Research Council (StG 309966).

## Author Contributions

V Hertlein: data curation and investigation.
H Flores-Romero: data curation and investigation.
KK Das: data curation and investigation.
S Fischer: investigation.
M Heunemann: data curation and investigation.
M Calleja-Felipe: investigation and methodology.
S Knafo: methodology.
K Hipp: investigation and methodology.
K Harter: methodology.
JC Fitzgerald: methodology.
AJ Garcia Saez: conceptualization, resources, supervision, funding ac-quisition, methodology, project administration, and writing—original draft, review, and editing.

## Conflict of Interest Statement

The authors declare that they have no conflict of interest.

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
