## [Reviewer comments · Life Science Alliance]

Novel BRET-based proximity biosensor for the study mitochondria-ER contact sites

Vanessa Hertlein, Hector Flores-Romero, Kushal K. Das, Sebastian Fischer, Michael Heunemann, Maria Calleja-Felipe, Shira Knafo, Katharina Hipp, Klaus Harter, Julia C. Fitzgerald and Ana J. García-Sáez

DOI: 10.26508/lsa.201900600

Corresponding author(s): Ana J. García-Sáez, University of Tübingen

Review timeline:

Submission Date:	2019-11-12
Editorial Decision:	2019-11-18
Revision Received:	2019-11-27
Accepted:	2019-11-27

Scientific Editor: Andrea Leibfried

Transaction Report:

Please note that the manuscript was previously reviewed at another journal and the reports were taken into account in the decision-making process at Life Science Alliance.

Reviewer Reports

Reviewer #1 Review

Comments to the Authors (Required):

The paper by Hertlein et al. provides a potentially useful description of a novel MAM sensor construct. This construct relies on BRET between ER-localized luciferase that can oxidize mitochondrial mVenus. At the moment, further control experiments must be produced to validate the construct. The concerns are serious, since ER and mitochondria are surrounded by redox nanodomains, which could seriously impede the functioning of MERLIN. The inability to show tighter MAM upon ER stress is another serious concern, since this effect has been shown by multiple labs. The standard knockdown of PACS-2 to show uncoupling is missing.

Specific points:

1. The BRET approach raises serious concerns. As shown by the Hajnoczky lab recently in Booth et al., mitochondria are surrounded by redox nanodomains. These could seriously impede the functioning of MERLIN. Indeed, the authors show that MERLIN with mitochondrial targeted luciferase is not efficient. Therefore, numerous additional controls must be performed to control for this possibility. I am listing some below.
2. Functioning mitochondria cause ROS. MERLIN must be tested in cells, where mitochondria are shut down. The best approach would be to do this in hypoxic cells. The authors should use both ER and mitochondria-targeted luciferase for these tests.
3. ER stress is a known factor that reinforces MAMs (see Bravo et al., 2011 and Csordas et al., 2006). However, the authors show the opposite with MERLIN (albeit with bortezomib that has not been tested before and at times, when this effect may no longer occur). An ER stress time course experiment using tunicamycin must be shown.
4. The authors use calnexin to target their probe to the MAM. Is this chaperone MAM-targeted? How? Can the authors interfere with this?
5. NAC is known to dissolve MAMs. What happens to MERLIN under this condition?
6. The SPLICS probe must be discussed in the introduction.
7. Mitofusin-2 KO results in high ROS. Is this effect behind the loss of signal?
8. PACS-2 KO or KD is the most reliable and undisputed way to detach ER from mitochondria. This should be put in relation to the PDZD8 and mitofusin-2 experiments (i.e., which one most detaches the organelles and reduces the signal).
9. The siRNA gels in Figure 3 are of very low quality and must be replaced by proper, clean signals.
10. STS is not useful for the studies, since it compresses cells. This is actually seen in the figure and could easily lead to artifacts.

Minor Points

1. The formatting of the references changes throughout the manuscript.
2. PDZD8 is related to ERMES, hence the identity of ERMES is no longer elusive in mammalian cells.

Reviewer #2 Review

Comments to the Authors (Required):

The article by Hertler and colleagues proposes a new sensor for mitochondria-ER contacts (MERCs) based on bioluminescence. The field of ER-mitochondria connections greatly expanded, yet tools to explore this interorganellar interface are scarce and often prone to artefacts, as correctly pointed out by the Authors. However, the advantage of a BRET sensor over a FRET probe like the one devised by the group of Hajnoczky and ameliorated by the group of Scorrano is unclear and unaddressed in this manuscript. The requirement for a manuscript to be considered competitive for this journal would be a comparison between the two probes in a setting where the BRET can be really advantageous (i.e., a HTS where FRET might be hampered by phototoxicity).

Indeed, the authors provide a naïve overview of the techniques available for ER-mitochondria proximity measurements and wrongly include the FRET probes in the category of the split GFP ones that can lead to artificial juxtaposition. The presence of a rapamycin dependent dimerization domain in the FRET probe by Hajnoczky allows the artificial dimerization and hence the maximization of the FRET signal (to be used as an internal reference for the efficiency of transfection, for example) but basically the FRET depends, like the BRET in their construct, on the proximity between the two fluorophores. Other probes, like the ones devised by the Schuldiner or by the Campbell groups (a non zippering ddGFP in this latter case) are simply not introduced, perhaps to reinforce the necessity for a BRET probe, but this is not a good service to the reader who should be able to get a factual overview of the field in the introduction. In conclusion, interorganellar biology is a booming field and tools to investigate it are much needed. However, this BRET probe is incremental over existing FRET probes to which it is not compared. Besides the report of the probe, the paper does not address a biological question and hence it is better suited in a specialty journal.

Additional aspects

1. The sentence "However, these methods also have drawbacks, most importantly because the establishment of artificial physical links between the ER and the mitochondrial membrane can affect the composition, dynamics, stability and regulation of the MERCs under investigation, thereby leading to artifacts. In addition, the establishment of this physical link between the two organelles is in many cases irreversible and limits their application to study MERCs dynamics." is not accurate, as mentioned above. While it holds true for dimerization based probes, FRET based probes are prone to zipper artifacts only upon addition of rapamycin, not in basal condition. Further, fixation of Rapamycin-treated cells in the plateau phase guarantees the possibility to measure the maximum achievable proximity between the 2 organelles in a certain cell type.
2. Perhaps the only concern of a FRET probe regards the range of distances that can be measured through FRET (but this depends on the linker region that connects the targeting sequences of each moiety to the fluorescent protein). The latter is, by the way, an issue that affects also the proposed BRET sensor.
3. Their method requires to treat cells as follows: "cells were washed with PBS, incubated with 5 μ M Coelenterazine h (Promega) in PBS for 10min in the dark". How could they exclude that this treatment is somehow altering MERCs, thus generating artefacts? MERCs must be measured eg by EM before and after Coelenterazine treatment
4. Can mTagBFP2: act as a donor for mVENUS, thus explaining the increase in the detected signal?

This is not verified in the paper

5. Authors state that Bortezomib, also known as PS-341, is a proteasome inhibitor that induces unfolded protein response (UPR) and ER stress. Different drugs inducing ER stress have been shown to increase mitochondria-ER contact sites (Bravo et al 2011). If the model reported by others is correct, they should have detected an increased, not a decreased, BRET signal. Again, authors fail to compare their BRET sensor with other approaches used to measure interorganellar proximity.

Alternatively, because PS341 is not commonly used to induce ER stress, they should have validated these results with other drugs.

6. Figure 2, how did they confirm the reported distances (3, 6, 12nM)? Immunogold EM images are not shown.

7. Fig3: quantification (ratio to actin) of the w.blot needs to be provided: actin seems to decrease during the time of siRNA treatment.

8. When measuring the effect of Mfn1, Mfn2 and Drp1 siRNA on BRET, several controls are missing:

- a. Reciprocal measurements of each dynamics protein upon silencing of the others
- b. Morphology of mitochondria/ER
- c. Targeting of the probe

9. When measuring the effect of starvation on BRET, several controls are missing:

- a. Levels of known tethers
- b. Morphology of mitochondria/ER
- c. Targeting of the probe

Reviewer #3 Review

Comments to the Authors (Required):

Quantitatively assessing organelle contacts in live cells is challenging. The advantage of using methods based on energy transfer (FRET or BRET) is that they do not generate or alter contacts. BRET has not previously been used to assess contacts and this study describes a BRET system to quantitatively measure ER-mitochondria contacts as a proof of principle. While the system has a good deal of promise, substantial additional work is necessary to demonstrate that it can quantitatively determine differences in contact in various conditions.

1. BRET signal is dependent on donor and acceptor expression levels, particularly when the acceptor is not in excess of the donor. Therefore, to meaningfully compare different conditions it is necessary to use cell lines that are stably transfected with the donor and acceptor fusions. It is also necessary to quantify how much of the fusions are in each cell line and to verify that localization of the fusions is not altered by addition of drugs and other conditions that affect cell growth.

2. How do the authors know that all the conditions they use do not reduce luciferase activity or Coelentrastazine uptake rather than directly affecting BRET?

3. It is not clear how $[mVen]/[rLuc]$ in Figure 2a,b was determined.

4. For the experiment in Figure 4b, it is necessary to know whether the expression level or localization of the fusions was altered at each time point. It is not clear what the different lines are.

5. The differences in contact shown in Figure 3 should be verified by another method, preferably EM.

1st Authors' Response to Reviewers

Reviewer #1 (Comments to the Authors (Required)):

The paper by Hertlein et al. provides a potentially useful description of a novel MAM sensor construct. This construct relies on BRET between ER-localized luciferase that can oxidize mitochondrial mVenus. At the moment, further control experiments must be produced to validate the construct. The concerns are serious, since ER and mitochondria are surrounded by redox nanodomains, which could seriously impede the functioning of MERLIN. The inability to show tighter MAM upon ER stress is another serious concern, since this effect has been shown by multiple labs. The standard knockdown of PACS-2 to show uncoupling is missing. We thank the reviewer for acknowledging the usefulness of our new MAM sensor. We have now performed a number of controls to address the reviewer's concerns, which we present below.

Specific points:

1. The BRET approach raises serious concerns. As shown by the Hajnoczky lab recently in Booth et al., mitochondria are surrounded by redox nanodomains. These could seriously impede the functioning of MERLIN. Indeed, the authors show that MERLIN with mitochondrial targeted luciferase is not efficient. Therefore, numerous additional controls must be performed to control for this possibility. I am listing some below.

There are different potential reasons that could explain why the efficiency of MERLIN at mitochondria and ER is different and we have extended now the discussion of this aspect in the text. We think the most likely could be due to different expression levels of Donor and Acceptor in the two organelles, which could be critical considering that BRET is highest when the donor acceptor ratio is 1:3. As the reviewer suggest, it is possible that the redox environment could affect the luciferase reaction, but taking into account the reviewer's comments we have now done additional controls to exclude this possibility (see below).

2. Functioning mitochondria cause ROS. MERLIN must be tested in cells, where mitochondria are shut down. The best approach would be to do this in hypoxic cells. The authors should use both ER and mitochondria-targeted luciferase for these tests.

Following the reviewer's suggestion, we have now tested MERLIN in hypoxic cells (new Figure 4 and S3). As shown in panel S3g, MEA-induced hypoxic conditions and ROS production, did not alter significantly the luciferase activity neither the MERLIN signal under the conditions tested and indeed, hypoxia had no significant effect on the BRET signal (Figure 4). These experiments were performed with mitochondrial-targeted luciferase because we used the newly generated cell line stably expressing MERLIN for more comparable conditions.

3. ER stress is a known factor that reinforces MAMs (see Bravo et al., 2011 and Csordas et al., 2006). However, the authors show the opposite with MERLIN (albeit with bortezomib that has not been tested before and at times, when this effect may no longer occur). An ER stress time course experiment using tunicamycin must be shown.

Following reviewer's suggestion, we have carried out tested the effect of tunicamycin in in the BRET signal of MERLIN-containing cell lines. As predicted, tunicamycin treatment dramatically increased the BRET signal, which is correlated with a decrease in the ER-mitochondria distance (new Figure 4 and S3c). Of note, the mechanism by which bortezomib and tunicamycin induce ER stress is not the same and this issue could perhaps explain the different effects observed at the concentration tested. We have now mentioned this in the text. See also response to reviewer 2.

4. The authors use calnexin to target their probe to the MAM. Is this chaperone MAM-targeted? How? Can the authors interfere with this?

Our ER construct contains the N-terminus of calnexin for targeting to the ER, where it can be detected with a homogeneous distribution under healthy conditions (see Figure 1). While it is not specifically targeted to the MAM, the identification of calnexin at MAMs previously in the literature (Myhill N. et al. Mol Biol Cell 2008; Lynes EM et al. EMBO J 2012)) and the fact that we detect it homogeneously distributed along the ER tubes support that it is also found at the MAM.

Of note, the same applies to the mitochondrial part of MERLIN. Besides their homogeneous distribution within the organelles, they only contribute to the BRET signal when they are in close proximity, considering that RET is extremely dependent on the distance R^6 and only efficient for distances lower than 10nm. This is the reason why we are careful to state along the manuscript that MERLIN measures distances between mitochondria and ER. By doing that, we propose that it serves as a proxy for MAM.

5. NAC is known to dissolve MAMs. What happens to MERLIN under this condition?

We thank the reviewer for this suggestion. We have now performed new experiments that test the effect of NAC in MERLIN-containing cell lines. At short times, the effect of NAC was not observable (data not shown), however after long-term incubation with NAC we detected a decrease in the BRET signal in a concentration dependent manner that is in agreement with a loosening of ER-mitochondria distances. These new data are now presented in Figure 3g.

6. The SPLICS probe must be discussed in the introduction.

We had already included the reference of SPLICS in the introduction as a tool that exploits split GFP to detect MAMs, and discussed its limitations, which also apply to other existing tools. Following the reviewer's suggestion, we have now mentioned the name more explicitly in this section.

7. Mitofusin-2 KO results in high ROS. Is this effect behind the loss of signal?

Considering that the data under hypoxic conditions show that ROS do not significantly affect the BRET signal of MERLIN, we argue that the loss of signal in Mitofusin-2 knock down is due to a decrease in the distances between ER and mitochondria.

8. PACS-2 KO or KD is the most reliable and undisputed way to detach ER from mitochondria. This should be put in relation to the PDZD8 and mitofusin-2 experiments (i.e., which one most detaches the organelles and reduces the signal).

We used PACS2 siRNA knockdown and obtained the expected result. However, despite several attempts we did not manage to obtain acceptable Western Blots to confirm the knockdown of PACS so we didn't include these results. Following the reviewer's suggestion, we now used instead NAC, which is also reducing the contacts between mitochondria and the ER. If the PACS experiment is still considered necessary, we would appreciate suggestions how to detect the protein by Western Blot.

9. The siRNA gels in Figure 3 are of very low quality and must be replaced by proper, clean signals.

Following the reviewer's suggestion, we present now a new gel in Figure 3b for siRNA knockdown of PDZD8. Moreover, we have now quantified the WB and we show the silencing of endogenous PDZD8 in a time dependent manner (Figure 3c).

10. STS is not useful for the studies, since it compresses cells. This is actually seen in the figure and could easily lead to artifacts.

We thank the reviewer for this comment. To control for this effect, we have now repeated the experiments in the presence of the pan caspase inhibitor QVAD, which blocks cellular contraction upon apoptosis induction by STS. As shown in the new Figure 3d, the increase in the BRET signal induced by STS is comparable also in the presence of QVAD. This control therefore discards cell shrinkage as a contributing factor in the BRET signal during apoptosis.

Minor Points

1. The formatting of the references changes throughout the manuscript.

We have now corrected this.

2. PDZD8 is related to ERMES, hence the identity of ERMES is no longer elusive in mammalian cells.

We agree with the reviewer and have now corrected this in the text.

Reviewer #2 (Comments to the Authors (Required)):

The article by Hertler and colleagues proposes a new sensor for mitochondria-ER contacts (MERCs) based on bioluminescence. The field of ER-mitochondria connections greatly expanded, yet tools to explore this interorganellar interface are scarce and often prone to artefacts, as correctly pointed out by the Authors. However, the advantage of a BRET sensor over a FRET probe like the one devised by the group of Hajnoczky and ameliorated by the group of Scorrano is unclear and unaddressed in this manuscript.

We agree with the reviewer that there is a need for tools to study interactions between organelles. To clarify the advantages of a BRET biosensor over FRET probes, we have now extended the comparison between them in the introduction and in the discussion. Basically, we find the following advantages of BRET over FRET:

-There is no dependence on the relative orientation between donor and acceptor, which increases the efficiency of the energy transfer.

-Because there is no need for illumination of the donor, there is no phototoxicity, which plays a role in live cell experiments of MERCs dynamics or in sensitive cell

types like neurons (see new Figure 4b,c, d) and as the reviewer mentions, in high-throughput screenings ([Figure for referees not shown in this Review Process File]).

-For the same reason, there are no cross-talk issues either and since all the signal comes from the energy transfer by BRET, the signal to noise ratio is increased, which becomes highly relevant for the practical use of the biosensor (see below).

The requirement for a manuscript to be considered competitive for this journal would be a comparison between the two probes in a setting where the BRET can be really advantageous (i.e., a HTS where FRET might be hampered by phototoxicity). We have developed the MERLIN system both in BRET and FRET formats, and report FLIM-FRET measurements in Figure 5. During the development of the tool, we have therefore extensively compared the use of FRET and BRET for detecting MERCs and found that in practice BRET works overall better, which is the reason why we focused on the BRET approach. In general, we find that the most important factor is the lower signal-to-noise ratio in the FRET measurements, which makes it more difficult to obtain quantitative results reproducibly. Indeed if one checks in the literature, the results reported for FRET probes suggest difficulties in detection (Figure3C shows FRET values close to 0 units before addition of rapamycin, which largely increases to around 300 FRET units upon dimerizer addition in Csordas et al. Mol Cell 2010; in the FEMP system by the Scorrano group all calculations – Fig2A and FigS2 - are shown with respect to the maximum FRET upon dimerizer addition and no absolute FRET values are provided, Naon et al. PNAS 2016), in agreement with our experience. We indeed think that the presence of the additional rapamycin-induced dimerization to set the maximum FRET signal is key to get a reference for the low FRET detected under normal conditions – which in our opinion poses a drawback that is overcome by MERLIN.

We agree with the reviewer that one of the advantages of MERLIN is its suitability for high-throughput screenings. Indeed, we have performed already a chemical screening of 10.000 compounds with MERLIN and successfully identified several compounds that increase/decrease the BRET signal, which we are currently validating. However we consider that these experiments are out of the scope of this work and will be part of a separate study.

[Figure for referees not shown in this Review Process File.]

In addition, we have also implemented MERLIN to measure distances between mitochondria and ER in living neuronal progenitors and differentiated neurons, which are very sensitive to phototoxicity and cannot be easily investigated by electron microscopy either. These data are now included in new Figures 4b,c,d. These results demonstrate an important additional advantage of MERLIN compared to other methods to detect MERCs, as requested by the reviewer.

Indeed, the authors provide a naïve overview of the techniques available for ER-mitochondria proximity measurements and wrongly include the FRET probes in the category of the split GFP ones that can lead to artificial juxtaposition. The presence of a rapamycin dependent dimerization domain in the FRET probe by Hajnoczky allows the artificial dimerization and hence the maximization of the FRET signal (to be used as an internal reference for the efficiency of transfection, for example) but basically the FRET depends, like the BRET in their construct, on the proximity between the two fluorophores. Other probes, like the ones devised by the Schuldiner or by the Campbell groups (a non zippering ddGFP in this latter case)

are simply not introduced, perhaps to reinforce the necessity for a BRET probe, but this is not a good service to the reader who should be able to get a factual overview of the field in the introduction.

Following the reviewer's suggestion, we have also expanded the introduction to explain better the FRET/dimerization sensor by Hajnoczky and to include additional references for the biosensors reported by the Schuldiner and Campbell groups, which are based on physical links between two proteins or protein fragments.

In conclusion, interorganellar biology is a booming field and tools to investigate it are much needed. However, this BRET probe is incremental over existing FRET probes to which it is not compared. Besides the report of the probe, the paper does not address a biological question and hence it is better suited in a specialty journal.

We agree with the reviewer that we do not address a biological question in our study, but would also like to point out that we have submitted our manuscript as a Tool, as it basically describes a new method. We hope that the additional data presented after the revision show better the advantages of MERLIN over existing tools.

Additional aspects

1. The sentence "However, these methods also have drawbacks, most importantly because the establishment of artificial physical links between the ER and the mitochondrial membrane can affect the composition, dynamics, stability and regulation of the MERCs under investigation, thereby leading to artifacts. In addition, the establishment of this physical link between the two organelles is in many cases irreversible and limits their application to study MERCs dynamics." is not accurate, as mentioned above. While it holds true for dimerization based probes, FRET based probes are prone to zipper artifacts only upon addition of rapamycin, not in basal condition. Further, fixation of Rapamycin-treated cells in the plateau phase guarantees the possibility to measure the maximum achievable proximity between the 2 organelles in a certain cell type.

See comments above.

2. Perhaps the only concern of a FRET probe regards the range of distances that can be measured through FRET (but this depends on the linker region that connects the targeting sequences of each moiety to the fluorescent protein). The latter is, by the way, an issue that affects also the proposed BRET sensor.

We agree with the reviewer that the distance between donor and acceptor is a factor that contributes to the efficiency of the energy transfer both in FRET and BRET. In MERLIN, we have devised a set of biosensor constructs with different linker lengths and selected the optimal linkers for measuring the distance between mitochondria and ER. In our case, the linker also has a helical folding, thereby allowing a better estimation of the actual linker length and avoiding unknown structural fluctuations associated with a theoretically flexible linker, as used in SPLICS.

3. Their method requires to treat cells as follows: "cells were washed with PBS, incubated with 5 μ M Coelenterazine h (Promega) in PBS for 10min in the dark". How could they exclude that this treatment is somehow altering MERCs, thus

generating artefacts? MERCs must be measured eg by EM before and after Coelenterazine treatment

Following the reviewer's suggestion, we have controlled for the effect of coelenterazine on the MERLIN signal. We found that coelenterazine H did not affect the MERCs because adding increasing concentrations of coelenterazine H to the assay did not alter the measured BRET ratios, only the measured values for RLuc and mVenus were increased. See new figure S3h.

4. Can mTagBFP2: act as a donor for mVENUS, thus explaining the increase in the detected signal? This is not verified in the paper

In those experiments there is no illumination source that could excite mTagBFP2 precisely because we are measuring BRET, so there is no possibility that mTagBFP2 is transferring energy to mVenus. We provide evidence for this in new Figure S3a.

5. Authors state that Bortezomib, also known as PS-341, is a proteasome inhibitor that induces unfolded protein response (UPR) and ER stress. Different drugs inducing ER stress have been shown to increase mitochondria-ER contact sites (Bravo et al 2011). If the model reported by others is correct, they should have detected an increased, not a decreased, BRET signal. Again, authors fail to compare their BRET sensor with other approaches used to measure interorganellar proximity. Alternatively, because PS341 is not commonly used to induce ER stress, they should have validated these results with other drugs.

We thank the reviewer for this comment. We have now studied the effect of ER stress on the MERLIN signal using tunicamycin and we detected an increase in the BRET signal in agreement with an increase in mitochondria-ER contact sites as expected (new Figure 4). See also response to reviewer 1.

Furthermore, we acknowledge the importance of comparing MERLIN with an alternative method to measure interorganellar proximity and we have now validated our measurements with electron microscopy. These new data are shown in Figure S3c. See also response to reviewer 3.

6. Figure 2, how did they confirm the reported distances (3, 6, 12nM)? Immunogold EM images are not shown.

These distances are theoretical, as mentioned in the text, and calculated based on the number of amino acids forming an alpha helix, which is an advantage of using a helical linker with known folding properties. See references in the text: (Kolossoff et al., 2008; Marqusee and Baldwin, 1987).

7. Fig3: quantification (ratio to actin) of the w.blot needs to be provided: actin seems to decrease during the time of siRNA treatment.

Following the reviewer's suggestion, we have now quantified the WB (new Figure 3c).

8. When measuring the effect of Mfn1, Mfn2 and Drp1 siRNA on BRET, several controls are missing:

- a. Reciprocal measurements of each dynamics protein upon silencing of the others
- b. Morphology of mitochondria/ER
- c. Targeting of the probe

We thank the reviewer for this comment. We have now confirmed that the morphology of mitochondria and ER are affected as expected by the silencing of the different proteins. We have also checked that the parts of the MERLIN biosensor still are properly targeted. See new Figure S6. We do not see why measuring each mitochondrial dynamics protein upon silencing the others is relevant for this study.

9. When measuring the effect of starvation on BRET, several controls are missing:
 - a. Levels of known tethers
 - b. Morphology of mitochondria/ER
 - c. Targeting of the probe

Following the reviewer's suggestion, we have estimated the levels of the mVenus part of MERLIN from the fluorescence values without substrate (new Figure S3a) and that RLuc activity is unchanged upon starvation (Figure S3g). We have also confirmed the morphology of the mitochondria and ER by EM (Figure S3c). We do not see why the levels of known tethers under starvation are of relevance for this study.

Reviewer #3 (Comments to the Authors (Required)):

Quantitatively assessing organelle contacts in live cells is challenging. The advantage of using methods based on energy transfer (FRET or BRET) is that they do not generate or alter contacts. BRET has not previously been used to assess contacts and this study describes a BRET system to quantitatively measure ER-mitochondria contacts as a proof of principle. While the system has a good deal of promise, substantial additional work is necessary to demonstrate that it can quantitatively determine differences in contact in various conditions. We thank the reviewer for acknowledging the interest of MERLIN.

1. BRET signal is dependent on donor and acceptor expression levels, particularly when the acceptor is not in excess of the donor. Therefore, to meaningfully compare different conditions it is necessary to use cell lines that are stably transfected with the donor and acceptor fusions. It is also necessary to quantify how much of the fusions are in each cell line and to verify that localization of the fusions is not altered by addition of drugs and other conditions that affect cell growth.

Following the reviewer's suggestion, we have now created cell lines stably expressing MERLIN. We have characterized them and successfully used them to test the different drugs and conditions. The new data are shown in Figures S3, 3d and 3g and 4a.

2. How do the authors know that all the conditions they use do not reduce luciferase activity or Coelentrastazine uptake rather than directly affecting BRET?

We thank the reviewer for this comment. Controls that the luciferase activity of RLuc is not affected by the treatment should be performed. We now discuss this in the manuscript and present it in Figure S3g.

3. It is not clear how [mVen]/[rLuc] in Figure 2a,b was determined.

The ratio [mVen]/[RLuc] corresponds to the concentration of transfected plasmid. The concentration of the plasmids could be used to determine the ratio because the

transfection was very robust. We obtained similar transfection levels throughout the different transfections and ratios. We verified this by microscopy counting the mVenus transfected cells to the whole cell population (Hoechst staining). For all the different ratios we had about 75% transfection efficiency. In addition, we

measured a linear increase in mVenus fluorescence for the increasing concentration of mVenus plasmid transfection (Figure R2 below).

Figure R2

4. For the experiment in Figure 4b, it is necessary to know whether the expression level or localization of the fusions was altered at each time point. It is not clear what the different lines are.

As the reviewer mentions, the localization and expression levels of the fusions in new Figure 3f remained unaltered as detected with microscopy, as shown in new Figure 3e for representative examples. The different lines correspond to changes in the BRET signal for each repetition of apoptosis induction in the cell population. We have now specified this in the figure caption.

5. The differences in contact shown in Figure 3 should be verified by another method, preferably EM.

We thank the reviewer for this comment. We have now validated the changes in mitochondria/ER distances detected with MERLIN using electron microscopy as an alternative approach. These data are shown in new Figure S3c. See also response to reviewer 2.

1st Editorial Decision

November 18, 2019

November 18, 2019

RE: Life Science Alliance Manuscript #LSA-2019-00600-T

Dear Dr. Garcia Saez,

Thank you for transferring your revised manuscript entitled "Novel BRET-based proximity biosensor for the study mitochondria-ER contact sites" to Life Science Alliance. Your manuscript was reviewed at another journal before and revised in response, and the editors transferred all versions, reviewer reports and your point-by-point response to us with your permission.

I have now assessed the revised version, and I am happy to say that I find your tool in principle suitable for publication here. A few amendments are still required before sending an official acceptance letter:

- The hypoxia treatment used to alter mitochondrial function and to then test BRET is lacking a control to show that mitochondrial function is indeed affected by the treatment => it would be good to add this control.
- I think the western blot in figure 3b is not ideal based on the actin control => please replace.
- Please explain in the figure/figure legend the grey versus dark lines in Fig S4.
- Please mention the statistical tests used whenever mentioning p-values in the figure legend.
- Please fix the point-by-point response - reference to newly added figures is sometimes mixed up for Fig S3 and S4.
- Please link your profile in our submission system to your ORCID iD, you should have received an email with instructions on how to do so.
- Please fill in all missing information in our submission system, including the author contributions.
- Please alter the running title slightly (too short).
- Please upload all figures as individual files, the legends (also S figure legends) should remain in the manuscript text.
- For supp. figures, you sometimes write fig. 2S instead of fig. S2
- Please add callouts to all figure panels for Fig S3 (currently only b, e, g)
- Please add a callout in the ms text to figure S5
- Please add callouts in the ms for Fig 4b-d
- Please provide the main ms text as docx file
- The inset in Fig5a is not correct for the zoomed portion shown, please fix
- The inset in Fig S3b is difficult to see, please change color. S3a displays two scale bars.

A. FINAL FILES:

-- High-resolution figure, supplementary figure and video files uploaded as individual files: See our detailed guidelines for preparing your production-ready images, <http://www.life-sciencealliance.org/authors>

B. MANUSCRIPT ORGANIZATION AND FORMATTING:

Full guidelines are available on our Instructions for Authors page, <http://www.life-sciencealliance.org/authors>

****Reviews, decision letters, and point-by-point responses associated with peer-review at Life Science Alliance will be published online, alongside the**

manuscript. If you do want to opt out of having the reviewer reports and your point-by-point responses displayed, please let us know immediately.**

Thank you for your attention to these final processing requirements. Please revise and format the manuscript and upload materials within 7 days. Thank you for this interesting contribution, we look forward to publishing your paper in Life Science Alliance.

Sincerely,

The hypoxia treatment used to alter mitochondrial function and to then test BRET is lacking a control to show that mitochondrial function is indeed affected by the treatment => it would be good to add this control.

We thank the reviewer for the comment, and following the suggestion we have now measured mitochondrial ROS production upon hypoxia treatment. As it is shown in the new figure S3g, the treatment with 25nM MEA, induces ROS production after 4h of incubation, measured by the change in the spectral properties (from red to green) of BODIPY, a ROS-sensitive dye.

- I think the western blot in figure 3b is not ideal based on the actin control => please replace.
 - Please explain in the figure/figure legend the grey versus dark lines in Fig S4.
 - Please mention the statistical tests used whenever mentioning p-values in the figure legend.
 - Please fix the point-by-point response - reference to newly added figures is sometimes mixed up for Fig S3 and S4.
 - Please link your profile in our submission system to your ORCID iD, you should have received an email with instructions on how to do so.
 - Please fill in all missing information in our submission system, including the author contributions.
 - Please alter the running title slightly (too short).
 - Please upload all figures as individual files, the legends (also S figure legends) should remain in the manuscript text.
 - For supp. figures, you sometimes write fig. 2S instead of fig. S2
- Please add callouts to all figure panels for Fig S3 (currently only b, e, g)
- Please add a callout in the ms text to figure S5
 - Please add callouts in the ms for Fig 4b-d
 - Please provide the main ms text as docx file
 - The inset in Fig5a is not correct for the zoomed portion shown, please fix
 - The inset in Fig S3b is difficult to see, please change color. S3a displays two scale bars.

We thank the reviewer for these comments and we have now corrected these aspects in the new version of the manuscript and the first point-by-point answer the reviewers.

1st Revision - Editorial Decision

November 27, 2019

November 27, 2019

RE: Life Science Alliance Manuscript #LSA-2019-00600-TR

Prof. Ana J. Garcia Saez
University of Tübingen
Hoppe-Seyler-Str. 4
Tuebingen 72076
Germany

Dear Dr. Garcia Saez,

Thank you for submitting your Research Article entitled "Novel BRET-based proximity biosensor for the study mitochondria-ER contact sites". I have re-assessed the further revised version of your paper now. I think the original concern of one of the reviewers regarding the siRNA treatment showing an effect only at a late time point is OK given that you did the BRET analysis at 48-72 hours post transfection and given the observed signal changes. I also appreciate the other introduced changes and it is a pleasure to let you know that your manuscript is now accepted for publication in Life Science Alliance. Congratulations on this interesting work.

DISTRIBUTION OF MATERIALS:

Authors are required to distribute freely any materials used in experiments published in Life Science Alliance. Authors are encouraged to deposit materials used in their studies to the appropriate repositories for distribution

to researchers.

Again, congratulations on a very nice paper. I hope you found the review process to be constructive and are pleased with how the manuscript was handled editorially. We look forward to future exciting submissions from your lab.

Sincerely,
